# Anthracyclins Increase PUFAs: Potential Implications in ER Stress and Cell Death

**DOI:** 10.3390/cells10051163

**Published:** 2021-05-11

**Authors:** David Balgoma, Fredrik Kullenberg, Carlemi Calitz, Maria Kopsida, Femke Heindryckx, Hans Lennernäs, Mikael Hedeland

**Affiliations:** 1Analytical Pharmaceutical Chemistry, Department of Medicinal Chemistry, Uppsala University, 751 23 Uppsala, Sweden; mikael.hedeland@ilk.uu.se; 2Translational Drug Development and Discovery, Department of Pharmaceutical Biosciences, Uppsala University, 751 23 Uppsala, Sweden; fredrik.kullenberg@farmbio.uu.se (F.K.); hans.lennernas@farmbio.uu.se (H.L.); 3Department of Medical Cell Biology, Uppsala University, 751 23 Uppsala, Sweden; carlemi.calitz@mcb.uu.se (C.C.); maria.kopsida@mcb.uu.se (M.K.); femke.heindryckx@mcb.uu.se (F.H.)

**Keywords:** hepatocellular carcinoma, lipidomics, plasmenyl, plasmanyl, plasmalogen, ferroptosis

## Abstract

Metabolic and personalized interventions in cancer treatment require a better understanding of the relationship between the induction of cell death and metabolism. Consequently, we treated three primary liver cancer cell lines with two anthracyclins (doxorubicin and idarubin) and studied the changes in the lipidome. We found that both anthracyclins in the three cell lines increased the levels of polyunsaturated fatty acids (PUFAs) and alkylacylglycerophosphoethanolamines (etherPEs) with PUFAs. As PUFAs and alkylacylglycerophospholipids with PUFAs are fundamental in lipid peroxidation during ferroptotic cell death, our results suggest supplementation with PUFAs and/or etherPEs with PUFAs as a potential general adjuvant of anthracyclins. In contrast, neither the markers of de novo lipogenesis nor cholesterol lipids presented the same trend in all cell lines and treatments. In agreement with previous research, this suggests that modulation of the metabolism of cholesterol could be considered a specific adjuvant of anthracyclins depending on the type of tumor and the individual. Finally, in agreement with previous research, we found a relationship across the different cell types between: (i) the change in endoplasmic reticulum (ER) stress, and (ii) the imbalance between PUFAs and cholesterol and saturated lipids. In the light of previous research, this imbalance partially explains the sensitivity to anthracyclins of the different cells. In conclusion, our results suggest that the modulation of different lipid metabolic pathways may be considered for generalized and personalized metabochemotherapies.

## 1. Introduction

All cancers are characterized by an inherent metabolic reprograming that promotes tumorigenesis by facilitating and enabling proliferation, metastasis, and resistance to therapies [1,2]. Therefore, metabolomics and lipidomics play key roles in unravelling the metabolic transformation in cancer [3] and cancer treatment [4]. Hepatocellular carcinoma (HCC) is one of the most common cancers worldwide, and it is known to cause profound modifications in lipid metabolism [5]. Among others, the malignant transformation of hepatocytes dysregulates the de novo lipogenesis [5]. This altered lipid metabolism is involved in rapid tumor growth and adaptation to the tumor microenvironment [6].

Clinicians have used anthracyclins, such as doxorubicin (DOX) and idarubicin (IDA), as chemotherapeutic agents for more than five decades. Anthracyclins intercalate into the nucleus and mitochondrial DNA and subsequently inhibit the synthesis of proteins and affect the redox state of the cell. Anthracyclins also act on the mitochondrial electron transport and convert oxygen into reactive oxygen species that may cause mitochondrial dysfunction, change the redox state, and induce lipid peroxidation [7]. However, the interplay between anthracyclins and the lipidome is not fully understood. Depending on the lipophilic and amphiphilic nature of the anthracyclins, the lipidome of the cell affects drug internalization, and, consequently, its effect [8]. For example, DOX-resistant MCF-7 cells present an enrichment of glycerophospholipids and cholesterol lipids [9]. The changes in the lipidome may affect the further internalization of the drug and the mechanism of action of the drug to induce cell death, in which endoplasmic reticulum (ER) stress plays a key role. This interplay among anthracyclin uptake, the lipidome, and ER stress may partially explain the different sensitivity to anthracyclins in both cancerous and non-transformed cells (e.g., cardiomyocytes). The sensitivity is conditioned by the mechanism of cell death induced by anthracyclins, which depends on the cell type and drug concentration [10]. Consequently, understanding this interplay in anthracyclin/lipidome/cell death might open the possibility to: (i) improve diagnosis by biomarkers of the tumor resistance to anthracyclins [11,12], and (ii) propose novel treatments that may potentiate and/or complement the metabolic transformations caused by anthracyclins to induce cell-death.

Currently, there are few studies about the effect of DOX on the lipidome of cancer cells and even less on the effect of other anthracyclins. In addition, different studies have reported that lipidic modulation affects ER stress, cell death, and cardiotoxicity induced by anthracyclins [2,13,14]. Consequently, more research about the effect of anthracyclins on the cellular tumor lipidome is needed. In this study, we aimed to characterize the lipidic metabolic transformation of cancer cells after exposure to two different anthracyclins. Consequently, three different primary liver cancer cell lines (HepG2, Huh7, and SNU449) were treated with DOX or the more lipophilic and potent IDA [8,15]. Subsequently, we investigated the regulation of the different lipids by their fold change and found that both anthracyclins increased the levels of polyunsaturated fatty acids (PUFAs) and alkylacylglycerophosphoethanolamines (etherPEs) with PUFAs in all cell lines (Scheme 1). The changes in cholesterol lipids were cell type-specific, as they increased in Huh7 and SNU449 cells but did not present a clear trend in HepG2. Similarly, the glycerolipids of saturated and monounsaturated fatty acids did not present a consistent trend for both treatments in both cell lines. Finally, we discussed the relationship between the lipidome, ER stress, and the sensitivity to anthracyclins. From a broad perspective, our results suggest the possibility of using PUFAs and/or etherPEs with PUFAs as general adjuvants of anthracyclins to build new metabochemotherapy treatments.

## 2. Materials and Methods

### 2.1. Cell Culture and Treatment. Murine Model

Three cell lines (HepG2 ATCC^®^ HB-8065™, SNU 449 ATCC^®^ CRL-2234™, and Huh7 from Dilruba Ahmed, Karolinska Institute, Stockholm, Sweden) were cultured at 37 °C with 5% CO_2_ and 95% humidity within a CO_2_ incubator. HepG2 and Huh7 were cultured in GlutaMAX™ supplemented, high glucose DMEM (31966047, ThermoFisher Scientific, Stockholm, Sweden) supplemented with 1% antibiotic antimycotic solution (A5955-100ML, Sigma-Aldrich, Darmstadt, Germany) and 10% fetal bovine serum (FBS) (10270106, ThermoFisher Scientific, Stockholm, Sweden) (cell culture media + FBS: CCMFed). SNU449 was cultured in GlutaMAX™ supplemented, RPMI (61870036, ThermoFisher Scientific, Stockholm, Sweden) medium supplemented with 1% antibiotic antimycotic solution and 10% FBS (cell culture media + FBS: CCMFed). Standard culture medium without supplemented FBS was used during starvation (CCMSM). Misidentification of all cell lines was checked at the Register of Misidentified Cell Lines [16]. For authentication, extracted DNA from all three cell lines was sent to Eurofins Genomics (Ebersberg, Germany) for cell line authentication using DNA and short tandem repeat profiles. Mycoplasma contamination was also tested.

Cells were seeded at a density of 4·× 10^6^ cells per T75 flask (75 cm^2^, 60 mL) and allowed to attach overnight. Prior to treatment, CCMFed was removed, and the cells washed with 10 mL of phosphate-buffered saline (PBS) (P4417-100TAB, Sigma, Stockholm, Sweden). To allow stabilization of the cell cycle, 15 mL CCMSM was added to each flask containing cells 2 h prior to treatment. Cells were treated for 48 h with either DOX or IDA in the form of a stock solution in DMSO containing 100 mM anthracyclin to achieve the concentrations in Table 1. For the respective controls, the amount of DMSO was matched to the anthracyclin treatment (vehicle controls). In all cases, the percentage of DMSO was between 0.03% and 0.0005%, which was at least 30 times below the limit at which DMSO presented cytotoxicity in our previous experiments (1% of DMSO, Kullenberg et al., submitted [17]). The exposure concentration of the anthracyclins was chosen to have cell mortality higher than 50% at 48 h of treatment. After 48 h, the cells were washed twice with 5 mL PBS. A volume of 5 mL PBS was added to each flask and the cells gently scraped using a cell scraper 30cm long, 2cm wide (99003, TPP, Nordic biolabs, Täby, Sweden). The resulting cell suspension was collected and the cells counted using a TC20™ Automated cell counter and dual-chamber counting slides (1450011, Bio Rad, Stockholm, Sweden). By using the histogram/gating option of the automated cell counter, the weighted mean diameter of the cells was determined for each cell culture flask. The cell suspension was subsequently centrifuged at 140 g for 5 min, the supernatant was removed, and the pellet was resuspended in 250 µL ice-cold MilliQ water and kept at −80 °C until analysis.

For ER stress imaging and measurement, cells were fixed for 10 min in 4% paraformaldehyde and stored at 4 °C. Paraformaldehyde-fixed cells were washed with tris-buffer saline (TBS) solution, and antigen retrieval was conducted at 95 °C in sodium citrate buffer for 45 min. Blocking was followed by an overnight incubation at 4 °C with primer antibodies. A time of 40 min of incubation was used for the secondary antibody (Rabbit anti-mouse Alexa Fluor-488 or donkey anti-rabbit Alexa Fluor-633), and cell nuclei were stained with Hoechst for 5 min. Images were taken using an inverted confocal microscope (LSM 700, Zeiss) using Plan-Apochromat 20× objectives and the Zen 2009 software Zen 2009 software (Zeiss, Oberkochen, Germany). The different channels of immunofluorescent images were merged using ImageJ software. Quantifications were conducted blindly with ImageJ software by conversion to binary images for each channel and automated detection of staining on thresholded images.

For transferrin staining, we used a mouse model of HCC. Five-week-old male sv129-mice were injected bi-weekly with 35 mg kg^−1^ diethylnitrosamine or equal volumes of saline. In this model, tumors occur after 25 weeks [18], after which, in this study, mice were treated with 2 mg kg^−1^ doxorubicin twice per week for a duration of 3 weeks. All methods were approved by the Uppsala Ethical Committee for Animal Experimentation (DNR 5.8.18-0089/2020). Murine liver biopsies were formalin fixed for 24 h and subsequently embedded in paraffin, after which they were cut into 8 μm sections. For immunohistochemical staining, five samples per experimental group were stained using the Rabbit Specific HRP/DAB (ABC) Detection IHC Kit (ab64261, Abcam, Cambridge, UK) according to the manufacturer’s guidelines. A polyclonal transferrin receptor antibody (PA5-27739, ThermoFisher, Göteborg, Sweden) was incubated in a 1/100 dilution for 2 h at 37 °C in a humidified chamber. Images were obtained with a Leica microscope, using a 10× objective, and image analyses was performed with Fiji ImageJ. The transferrin receptor-positive staining was extracted using the color deconvolution plugin [19] and quantification of the thresholded images.

### 2.2. Lipidomics Analysis and Data Pre-Treatment

After randomization of the order of sample preparation, 100 µL of acetonitrile/isopropanol 50:50 was added to 10 µL of sample. Proteins were precipitated overnight at −20 °C, and the supernatant was recovered after centrifugation (18,000 RCF, 10 min). A quality control sample was built by pooling 10 µL from every extract. The samples were analyzed as in Balgoma et al. [20]. The samples were injected in a randomized way on an Acquity UPLC hyphenated to a Synapt G2 Q-ToF (Waters, Manchester, UK) with electrospray ionization. The samples were analyzed in both positive and negative modes. The quality control pool was injected every five injections of samples.

Lipids were identified as in Balgoma et al. [20] by the m/z of their adducts and their patterns of fragmentation. Briefly, a first level of identification was performed by the information in MS mode (*m*/*z*), yielding the family of the lipid, the adduct, the number of total carbons, and the number of total unsaturations. In all cases, the maximum absolute deviance of *m*/*z* for all adducts was 10 ppm (Appendix A). For glycerophospholipids, when more than one fatty acid was possible and the fragmentation signal allowed it, the main combination of fatty acids of the lipid was performed by the fragmentation patterns, as reported in the literature [21,22]. For fatty acids, different isomers were separated chromatographically. By prior knowledge, the most abundant isomer was assigned to the most abundant fatty acid in mammalian cells (e.g., the most abundant FA(18:1) isomer was assigned to oleic acid, FA(18:1n-9).

In total, we detected 451 species of lipid peaks corresponding to free fatty acids (FA), lysophosphatidylcholine (LPC), phosphatidylcholine (PC), alkylacylglycerophospholipids of choline (etherPC), lysophosphatidylethanolamine (LPE), phosphatidylethanolamine (PE), etherPE, phosphatidylglycerol (PG), lysophosphatidylinositol (LPI), phosphatidylinositol (PI), lysophosphatidylserine (LPS), phosphatidylserine (PS), diacylglycerols (DG), TG, free sterols, cholesteryl esters (CE), ceramides (Cer), and sphingomyelin (SM). Due to the number of lipids, only the key species by their biological meaning are represented in the main text. The identification and the fold changes for all lipid species are reported in Appendix A.

### 2.3. Data Analysis and Biomedical Interpretation

We addressed the concerns about using *p*-values and “statistically significant” discoveries exposed by researchers, statisticians, and the American Statistical Association [23,24,25]. Consequently, we analyzed the changes in the lipidome by using fold changes (with confidence intervals) between the treatments and the control [26].

Cell size raw data consisted of the number of cells in a certain range (e.g., 3.1 million cells between 4 and 9 µm). To obtain the average of the cell size, we calculated the weighted mean: (1) for every interval, the central value was multiplied by the number of cells in the interval (in our example, 3.1 million cells × 6.5 µm); (2) we calculated the mean by summing the values in step 1 by the total number of cells. For the three culture replicates, the variability was calculated by the standard error of the mean.

Regarding lipidomics data, the areas of the peaks were normalized by the number of cells extracted. The changes in the lipidome were studied by the relative changes (increase/decrease) between the cells treated with anthracyclins and the cells treated with their respective controls (vehicle). These relative changes were quantified by the log (fold change), i.e., the natural logarithm of the ratio of the average of the normalized signal of the treated group divided by the average of the normalized signal of the control group (vehicle). The confidence interval of the log (fold change) was determined by 10,000 bootstrap resampling simulations. The limits of the intervals of confidence were selected by the percentiles 2.5% and 97.5% of the simulations; the central measurement was characterized by the mean of the simulations. The values under the limit of detection (left-censored) were imputed by fitting the peak areas of a lipid to a normal distribution and random sampling of this distribution below the minimum detected signal. Fold changes with more than one value under the limit of detection in any group were discarded.

Our lipidomics study was non-hypothesis-driven [27,28]. Aiming at a broad audience, we discussed our results in the biomedical context in an inductive framework. Consequently, in order to interpret our results, we compared our results with previous research.

### 2.4. Software and Graphical Material

Waters’ raw data files were transformed into .CDF format by Databride (Masslynx 4.1, Waters, Manchester, UK). Mass spectrometric data were pretreated with packages mzR 2.22.0 (https://www.bioconductor.org, accessed on 11 May 2021) and XCMS 3.10.2 (https://www.bioconductor.org, accessed on 11 May 2021) in R 4.0.3 “Bunny-Wunnies Freak Out” (https://cran.r-project.org/, accessed on 11 May 2021). Graphical material was generated with R (packages forestplot 1.10 and ggplot2 3.3.2, https://cran.r-project.org/, accessed on 11 May 2021) and processed with Inskape 1.0.1 (https://inkscape.org/, accessed on 11 May 2021) and Gimp 2.10.22 (https://www.gimp.org/, accessed on 11 May 2021). Microscopy images were obtained in Zen 2009 software (Zeiss, Germany) and further quantified and exported using ImageJ software (https://imagej.nih.gov/ij/, accessed on 11 May 2021). The endomembrane cell system in the graphical abstract comes from an image in Wikipedia with the following statement: “This work has been released into the public domain by its author, LadyofHats (Mariana Ruiz Villarreal). This applies worldwide”.

## 3. Results

### 3.1. Number and Size of the Recovered Cells

As expected, DOX and IDA induced a strong reduction in cell number when compared with the vehicle control. The respective vehicle controls did not induce cell death nor proliferation (Figure 1). In all cases, anthracyclins induced a reduction in cell number higher than 50%, when compared with the vehicle.

Regarding the cell size, the vehicle controls did not change the diameter of the cells when compared with the untreated control (Figure 2). The treatment with anthracyclins did not change the cell volume for HepG2 nor Huh7. In contrast, we observed an increased cell diameter in DOX- and IDA-treated SNU449 cells (Figure 2).

### 3.2. Anthracyclins Increase etherPEs in All Cell Types

To estimate the regulation of the total amount of lipids per family, we summed the normalized signal of every lipid family and calculated the logarithm of the fold change with respect to the vehicle-treated cells (Figure 3).

In general, the different families of lipids presented no clear trend for all lipids for the three cell types. Regarding the ensemble of the lipid families for every cell type, the lipidome showed a trend to increase in SNU449, especially with DOX (Figure 3).

Regarding common trends for both treatments and the three cell lines, we observed an increase in etherPEs. While most families of lipids increased in SNU449, the fold change of etherPEs was the highest, together with cholesterol lipids. This behavior points to a common and general metabolic effect of anthracyclins in etherPEs beyond the specificities of the metabolism of every cell line.

Regarding disparate trends for the different cell lines, cholesterol lipids (sterols and CEs) presented a trend to increase in Huh7 and SNU449, but not in HepG2. PGs presented a trend to increase in HepG2 and Huh7, but not a clear trend in SNU449. PSs and the sphingolipids (Cer and SM) presented a trend to decrease in HepG2, no clear trend in Huh7, and a trend to increase in SNU449. As highlighted before, the increase in these lipids in SNU449 was in a context of a general increase in lipids in this cell line. Furthermore, Huh7 treated with IDA presented a general trend to increase.

### 3.3. Anthracyclins Increase All etherPEs with PUFAs, but Not etherPC

To study which lipid pathways were responsible for the upregulation of etherPEs in Figure 3, we present the fold change of the individual species of alkylacylglycerophospholipids (etherGLs) and their fatty acid composition (Figure 4).

We only detected one species of etherPC, PC(O-16:0/18:1). This species did not present a clear trend for all cells and treatments (Figure 4). Regarding etherPEs, we detected six different species and all contained PUFAs. They corresponded to plasmanyl (ether bond) or plasmenyl (vinyl ether bond, Scheme 1), but their mass spectrometric characterization did not allow distinguishing these possibilities from each other. The fold change in etherPEs presented a common trend to increase in the three cell lines with both anthracyclins. We conclude that the upregulation of etherPEs with PUFAs (but not all etherGLs) was a general metabolic trait of anthracyclins.

### 3.4. Anthracyclins Increase Free PUFAs

To study if the availability of free PUFAs may partially control the increase in etherPEs with PUFAs, we investigated the fold change of FAs.

Despite the different regulation of different fatty acids, FA(20:4) and FA(20:5) showed a clear trend to increase in all cell lines treated with either DOX or IDA (Figure 5). Other PUFAs, such as FA(20:3) and FA(22:6), also presented a trend to increase, but to a lower degree, especially in SNU449. This behavior suggests that free PUFAs are more available in all anthracyclin-treated cells.

### 3.5. Anthracyclins Increase Other Glycerolipids with PUFAs

Radiolabeling experiments have shown that the glycerophospholipids of choline and ethanolamine contain most PUFAs [29]. Consequently, we studied the levels of glycerolipids with PUFAs to investigate if a net release of PUFAs would explain the increase in free PUFAs. We observed that different molecular species presented different regulations (Figure 6), which might be explained by the different substrate selectivity in the enzymes involved in the liberation and incorporation of PUFAs into glycerolipids [30]. However, the major species tended to increase in all cells with all treatments. This was the case of PC(16:0/20:4), PC(18:1/20:4), PE(18:0/20:4), PI(18:0/20:4), PC(16:0/20:5), PC(18:0/20:5), and PC(18:1/20:5) (Figure 6). Taken together, these upregulations suggest that the increase in free PUFAs (Figure 5) was not due to a net release of PUFAs from glycerolipids.

Interestingly, FA(22:6) (Scheme 1) might present a different regulation than other PUFAs, as it is produced from FA(24:6) by peroxisomal β-oxidation [31]. The major species of glycerolipids with FA(22:6) in PC and PE did not present a clear trend to decrease following anthracyclin treatment in any of the cell lines (Figure 6). Some minor species, such as PG(22:6/22:6) and TG(18:0/18:1/22:6), presented a trend to increase in treated HepG2 and Huh7. Nevertheless, these species did not present a clear increase in SNU449. In summary, the data do not suggest a general decrease for all cell lines of neither free nor esterified FA(22:6) (Figure 5 and Figure 6).

### 3.6. Regulation of Glycerolipids with Saturated and Monounsaturated Fatty Acids

Glycerolipids with FA(16:0), FA(16:1), FA(18:0), and FA(18:1) are markers of LXR and SREBP pathways for de novo lipogenesis [32]. To evaluate if anthracyclins affect the de novo lipogenesis in the three cell lines, we studied the changes in these lipid species (Figure 7).

Despite the specific decrease in species such as PC(14:0/16:1) in HepG2, the general overview of the main glycerolipid species with saturated and monounsaturated fatty acids did not show any clear trend. Only in Huh7 treated with IDA presented an increase in these species, except for TG(18:0/18:0/18:1). This suggests that anthracyclins do not affect LXRs’ nor SREBPs’ de novo lipogenesis in a general way, but it could depend on the cell type–treatment combination.

### 3.7. DOX Increases Markers of ER Stress

To study ER stress, we determined the levels of DNA damage-inducible transcript 3 (also known as C/EBP homologous protein, CHOP) and binding immunoglobulin protein (BIP) through immunocytochemistry (Figure 8). These two markers increased after DOX treatment, which confirms that anthracyclins induce ER stress. The increase was stronger for HepG2 and Huh7 than for SNU449.

### 3.8. DOX Induces the Expression of Transferrin Receptor in the Liver of an HCC Murine Model

To study the expression of markers of ferroptosis after treatment with DOX, we measured the expression of the transferrin receptor in the liver of a murine model of HCC (Figure 9) [33]. In comparison with the control of HCC, the expression of the transferrin receptor increased with the treatment with DOX. This suggests that DOX induces ferroptosis cell death in HCC.

## 4. Discussion

Here, we show that anthracyclins elicit the upregulation of PUFAs and etherPEs with PUFAs in different primary liver cancer cells. By using DOX and IDA in three different cell lines, we present new evidence that this is a general effect of anthracyclins on the metabolism of cancer cells. We also present evidence that the regulation of cholesterol lipids and de novo lipogenesis by anthracyclins is cell type-dependent for three different cell lines of primary liver cancer. As etherPEs with PUFAs are involved in programmed cell death by ferroptosis [34] and statins have been suggested as anthracyclin adjuvants [35], this metabolic characterization may help to design general and personalized metabochemotherapies for improved cancer treatments.

### 4.1. Regulation of the De Novo Lipogenesis of Fatty Acids, Glycerolipids, and Cholesterol

LXRs and SREBPs control the expression of the enzymes responsible for the de novo lipogenesis of fatty acids, glycerolipids, and cholesterol [36]. The increase in de novo lipogenesis of fatty acids and glycerolipids is characterized by the upregulation of glycerolipids with saturated and monounsaturated fatty acids [32]. In this context, we did not observe a general up- or downregulation of these glycerolipids, which suggests that anthracyclins do not modify the de novo synthesis of fatty acids in general (Figure 7). Regarding the regulation of cholesterol, we observed a trend to upregulation of cholesterol lipids in Huh7 and SNU449 (Figure 3). However, this trend was not clear for HepG2. Consequently, our data do not suggest a general trend for the de novo lipogenesis of neither fatty acids nor cholesterol. This agrees with previous studies, which did not find a clear trend of upregulation or downregulation for the LXR/SREPB pathways [37,38,39]. Our interpretation may be confounded by a putative reduction in beta-oxidation in the mitochondria and a subsequent accumulation of fatty acids. Nevertheless, considering that these lipids are consistently upregulated in different diseases with dysregulated lipogenesis [32], it seems plausible to consider them as biomarkers of the LXR/SREPB pathways.

Considering our results and previous research together, we conclude that the up- or downregulation of the LXR/SREBP-mediated de novo lipogenesis of fatty acids, glycerolipids, and cholesterol was not a general effect of anthracyclins. Nevertheless, as we saw different regulations for three different lines of the same type of cancer, it could depend on the cell type, tumor, or even individual genotypes and phenotypes.

### 4.2. Regulation of the Increase in etherPEs with PUFAs and PUFAs by Anthracyclins

The synthesis of etherGLs is initiated in the peroxisome and completed in the ER, where the activity of alcohol reductase FAR1 is the limiting step (Scheme 2) [40,41,42]. The amount of etherGLs is regulated by: (i) the downregulation of FAR1 by the increase in etherGLs in the inner leaflet of the plasma membrane, and (ii) the lysoplasmalogenase activity [43]. While we observed an upregulation of etherPEs with PUFAs in all cells and treatments (Figure 4), we detected an etherPC, PC(O-16:0/18:1), that did not present a consistent increase in all treatments and cells. This differential behavior suggests that the upregulation of etherPEs with PUFAs was independent of: (i) a putative increase in the synthesis of 1-O-alkyldihydroxyacetone phosphate in the peroxisome (Scheme 2), or (ii) a putative decrease in the lysoplasmalogenase activity. Free PUFAs increased in a general way. Consequently, it seems plausible that the availability of PUFAs was the metabolic regulation responsible for the increase in etherPEs with PUFAs.

The levels of free PUFAs could be increased by three different factors: (i) the increase in their synthesis by elongation and desaturation of essential linoleic and linolenic acids [45]; (ii) the balance between their liberation from and incorporation into glycerolipids [29,30,47,48]; and/or (iii) the downregulation of their degradation by peroxisomal β-oxidation [40]. Consequently, we analyzed the levels of PUFAs in different glycerolipids to discuss the potential involvement of these three processes. On the one hand, the general trend to upregulation of FA(20:4) and FA(20:5) in glycerolipids (Figure 6) suggests that their net release is not responsible for the increase in their free form. On the other hand, their degradation is initiated in the peroxisome by β-oxidation. Interestingly, docosahexaenoic acid, FA(22:6), is not synthesized by elongation and desaturation in the ER, but by β-oxidation of FA(24:6) in the peroxisome (Scheme 2) [49]. Consequently, the decrease in free and esterified FA(22:6) would be indicative of the downregulation of peroxisomal β-oxidation and, consequently, of the degradation of PUFAs. However, we did not observe a general downregulation of FA(22:6), neither free nor esterified. In fact, we rather observed a trend to increase in specific lipid species in HepG2 and Huh7 (Figure 6). The behavior of FA(22:6) suggests that neither the activity of peroxisomal β-oxidation nor the degradation of PUFAs was downregulated as a general characteristic of anthracyclins.

We conclude that the observation of the ensemble of free PUFAs and glycerolipids with PUFAs suggests the upregulation of the elongation and desaturation of linoleic and linolenic acids as responsible for the increase in free PUFAs and etherPEs with PUFAs (Scheme 2).

### 4.3. Role of PUFAS and etherPEs with PUFAs: Potential Hallmark of Programmed Cell Death by Ferroptosis

Regarding cell death by anthracyclins, PUFAs and etherGLs with PUFAs have been associated with lipid peroxidation and ferroptosis [34,50,51,52]. Ferroptosis, a term for a non-apoptotic cell death introduced in 2012, is characterized by the intracellular accumulation of lipid hydroperoxides [53,54]. Lipid peroxides yield aldehydes that react with intracellular and membrane proteins, compromising their function. To achieve lipid peroxidation, ferroptosis requires changes in a plethora of pathways, such as iron, glutaminolysis, and cholesterol and mevalonate pathways [54]. Furthermore, ferroptosis does not imply modifications in the morphology of the cell, except for the mitochondria [55].

In the context of lipid peroxidation and ferroptosis, first, our analysis about the regulation of PUFAs indicates that their increase was due to the upregulation of elongation and desaturation (Section 4.2, Scheme 2). In this process, Δ6-fatty acid desaturase 2 (FADS2) is necessary for the synthesis of PUFAs [56]. Interestingly, the knockdown of FADS2 presents attenuated lipid peroxidation in Huh7 cells [57]. Consequently, the upregulated elongation and desaturation to yield PUFAs (Section 4.2) are consistent with the increase in lipid peroxidation and ferroptosis.

Second, we found a general increase in etherPEs with PUFAs. It is known that: (i) glycerophospholipids of ethanolamine with PUFAs are necessary for ferroptosis [58]; and (ii) etherGLs with PUFAs present a critical contribution to ferroptosis [34]. It is plausible that the increase in etherPEs with PUFAs can be considered a hallmark of ferroptosis in primary liver cancer treated with anthracyclins.

Finally, regarding the morphology, we found that HepG2 and Huh7 did not change their size upon anthracyclin treatment. However, SNU449 increased in size with both DOX and IDA. Furthermore, we observed an increase in PUFAs and etherPEs with PUFAs in SNU449, but in the context of a general increase in lipids (Figure 3). EtherPEs with PUFAs presented a slightly higher fold change than other lipids in SNU449 (Figure 3), but the overall change of the lipidome and the increase in size suggest that the enrichment of lipids with PUFAs was diluted in the membranes. Consequently, it seems plausible that lipid peroxidation in SNU449 was quenched.

Considering the discussion in the three previous paragraphs, we conclude that the changes in the lipidome and the cell size suggest that ferroptosis may drive the cell death for HepG2 and Huh7. However, other mechanisms apart from lipid peroxidation and ferroptosis may drive the cell death for SNU449. This is consistent with previous studies, as the mechanism of cell death depends on the concentration of anthracyclins and the cell type [10]. To test the ferroptosis hypothesis suggested by our non-hypothesis-driven lipidomics study, we measured the expression of the transferrin receptor in a murine model of HCC treated with DOX (Figure 9). The strong increase in transferrin receptor strengthens our hypothesis that anthracyclins entail ferroptotic cell death.

### 4.4. Interplay Among the Lipidome, Anthracyclin Uptake, ER Stress, and Sensitivity to Anthracyclins

The metabolome, the lipidome, the proteome, and the transcriptome affect anthracyclin uptake and ER stress [59,60]. Anthracyclins themselves affect the-omes, which further modifies the uptake of the drug and ER stress. The outcome of this crossed interplay is the different mechanisms of cell death and hence the sensitivity of a cell/tumor type to anthracyclins [10]. Of special interest in this interplay is the lipidome, as it conditions: (i) drug uptake (drug—membrane interaction and membrane fluidity), (ii) ER stress (ratio PC/PE, balance of PUFAs versus cholesterol and saturated lipids), and (iii) the type of cell death (lipids with PUFAs in ferroptosis) (Scheme 3).

Regarding anthracyclin uptake, the lipid membrane composition determines its fluidity and drug membrane permeation, as well as activation of drug carrier-mediated efflux mechanisms related to multidrug resistance [8,61]. We have found that the intracellular uptake ratio of DOX was one order of magnitude higher in HepG2 or Huh7 than in SNU449 (HepG2 > Huh7 >> SNU449, Kullenberg et al., submitted [17]). Interestingly, in this study, PGs presented a remarkable increase in HepG2 and Huh7 with both anthracyclins (Figure 3). PGs are negatively charged and interact in the membrane with positively charged DOX and IDA. Other lipids also affect the properties of the membranes regarding drug uptake: (i) the levels of other negatively charged lipids (PIs, PSs); (ii) zwitterionic lipids (PCs, PEs) [62]; and fluidity (cholesterol and saturation of fatty acids in the glycerolipids). Our study was limited to three cell lines, which prevents the discussion of these multiple changes. Nevertheless, in the light of previous research, we speculate that the strong enrichment of PGs in the membranes of HepG2 and Huh7 (Figure 3) may play a role in favoring anthracyclin uptake.

ER stress plays an essential role in apoptosis and ferroptosis [63], as well as in promoting tumorigenesis in HCC [64,65]. In addition, ER stress mediates drug resistance to chemotherapeutic agents [66]. There is a crosstalk between ER stress pathways and the lipidome [67]. In this crosstalk, more than the composition of a specific type of lipid, it is the balance among different types of lipids that is associated with ER stress. For example, the imbalance between lipids with PUFAs versus saturated and cholesterol provokes ER stress [59,60]. In this context, we observed that: (i) membrane lipids with PUFAs increased in general in each of the three cell lines (Section 4.2), (ii) membrane lipids with saturated and monounsaturated fatty acids did not present any clear trend (Section 4.1), and (iii) cholesterol lipids increased in Huh7 and SNU449, but not in HepG2 (Figure 3). This ensemble of observations suggests a higher imbalance in HepG2 cells than in SNU449 (Scheme 3), which, in the light of previous research, partially explains the stronger increase in ER stress with DOX in HepG2 cells than in SNU449 (Figure 8). Many other factors beyond the scope of this study affect ER stress [67], which may explain why Huh7 did not present a strong imbalance between lipids with PUFAs and sterols but presented a strong increase in ER stress (Figure 8B).

Considering the ensemble data about the lipidome, drug uptake, and ER stress (Scheme 3), the observed changes may contribute to explaining why SNU449 is one of the most resistant cell lines to anthracyclins [68]. In comparison with HepG2 and Huh7, SNU449 presented: (i) an increase in size (Figure 2), (ii) a general increase in the lipidome (Figure 3), (iii) a weaker increase in ER stress makers (Figure 8). Despite the fact that other factors (metabolome, proteome, transcriptome) also take part in cell death, this differential behavior in the context of previous research suggests a key role of lipids in cell death. In this role, more than the change in one lipid or lipid family, it is the ensemble of lipids which seems involved in the resistance to anthracyclins.

### 4.5. Perspectives in Metabochemotherapy and Anthracyclins

It is known that PUFAs potentiate the effect of DOX in vitro and in animal models [69,70]. This effect might be general, as it would potentiate the lipid peroxidation in ferroptosis induced by anthracyclins. However, potentiating ferroptosis may increase the cardiotoxicity of anthracyclins [71]. Paradoxically, it has been reported that PUFAs may have a cardioprotective effect in anthracyclin chemotherapy [72,73]. This suggests that PUFAs may present a therapeutic window. In this window, PUFAs may potentiate anthracyclin-induced ferroptosis in tumor cells and, in parallel, minimize cardiotoxicity. Similarly, the administration of anthracyclins in bilayers with etherPEs with PUFAs may also present a therapeutic window. Our results suggest that this would be the case for tumors with cells with lipid metabolism similar to HepG2 and Huh7. As discussed before, ferroptosis might not be the only suggested mechanism of death for SNU449. Nevertheless, we speculate that the treatment with PUFAs and/or etherPEs with PUFAs might drive tumors with cells such as SNU449 into the path of lipid peroxidation and ferroptosis. In this perspective, PUFAs and etherPEs with PUFAs could be considered general adjuvants of anthracyclins.

Another suggested adjuvant of anthracyclins is the inhibition of cholesterol formation by statins [74]. Our results suggest that the effect of anthracyclins on cholesterol lipids was specific for each of the three cell lines of the same type of tumor. The fact that having a different behavior for three cell lines of the same type of tumor indicates that the modulation of the cholesterol pathway would be of interest depending on both the cancer type and the patient. This agrees with the previous literature, as statins present synergizing and protective effects in different models [35,75,76,77]. Furthermore, the adjuvant effect of statins is not clear in patients [74]. An explanation would be that the adjuvant effect of statins might depend on the patient pharmacokinetics, pharmacodynamics, and cholesterol metabolism. Consequently, while statins do not seem to be a general adjuvant of anthracyclins, it seems plausible that they could be used for specific tumors and in personalized metabochemotherapy.

## 5. Conclusions

To the best of our knowledge, here, we showed, for the first time, that the upregulation of free PUFAs and etherPEs with PUFAs is a general trait of anthracyclins. According to the literature, this increase is pro-ferroptotic by promoting lipid peroxidation, but other metabolic changes might counteract this type of cell death. This is the case of cholesterol lipids and saturated and monounsaturated glycerolipids, which are markers of the de novo lipogenesis in the LXR/SREBP pathways. PGs, which are negatively charged at physiological and pathophysiological pH, presented a strong increase in HepG2 and Huh7. This increase may play a key role in the higher uptake ratio of DOX of these cell lines. We found that these lipids showed disparate behavior depending on the cell type after the treatment with anthracyclins. These specificities partially explain the different sensitivity to anthracyclins by the effect of lipids on passive diffusion of the drugs, their metabolites, membrane fluidity, ER stress, and the mechanism(s) of cell death.

In conclusion, the behavior of the lipidome in the three cell lines suggests that: (i) PUFAs or PUFA-containing lipids may potentiate the effect of anthracyclins in a general way, and (ii) statins may potentiate or attenuate the effect of anthracyclins in a cancer/patient-specific way. Further research in animal models and patients is warranted to establish the possibility of transferring these observations from bench to bedside.

## Data Availability

The fold changes and their confidence intervals are available at Appendix A.

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
