# Peer review of "Anthracyclins Increase PUFAs: Potential Implications in ER Stress and Cell Death"

_cells, 2021, doi:10.3390/cells10051163_

Round 1

Reviewer 1 Report

David Balgoma et al in this work analyzed the lipidomics in the liver cancer cell lines (HepG2, Huh7 and SNU449) in the absence and presence of two anthracyclins (doxorubicin and idarubin) and found that the levels of polyunsaturated fatty acids (PUFAs) and alkylacylglycerophosphoethanolamines (etherPEs) with PUFAs were increased after the anthracyclins treatment, which was accompanied with

endoplasmic reticulum stress. Since PUFAs are fundamental in lipid peroxidation during ferroptosis, the authors suggest supplementation with PUFAs and/or etherPEs with PUFAs as a potential general adjuvant of anthracyclins. The drawback of this manuscript is the lack of mechanism study, without any ferroptosis analysis or how the PUFAs were increased. Is the production of PUFAs mediated by SCD1? The authors think PUFAs are the triggers of ferroptosis, but doxorubicin and idarubin only induce one type of cell death or mixed types of cell death. Common idea is that PUFAs are good lipids, how does the story here explain that?  Other minor comments:

1. The data presented without statistics, just show the trend of changes.

Author Response

We kindly thank Reviewer 1 for the comments. We have addressed them:

The drawback of this manuscript is the lack of mechanism study, without any ferroptosis analysis or how the PUFAs were increased. Is the production of PUFAs mediated by SCD1?

A deep mechanistic study would require the use of pharmacological interventions on the cell cultures. From a lipidomics point of view, this would require the use of isotopically labeled lipids. The primary aim of our study was to provide an untargeted lipidomics analyses to discuss the results in a broad perspective. Consequently, we focused on the interpretation of our untargeted data on the light of the mechanisms described by previous researchers.

The increase of PUFAs can be addressed to a higher activity of elongation and desaturation of essential linoleic and linolenic acids (Scheme 2). In higher animals, the enzymes for the elongation and desaturation of linoleic and linolenic acid are responsible for the synthesis of PUFAs (except FA(22:6)). To the best of our knowledge, SCD1 is not involved in the synthesis of PUFAs in animals. The enzymes involved in the desaturation to yield PUFAs are FADS1 and FADS2.

The authors think PUFAs are the triggers of ferroptosis, but doxorubicin and idarubin only induce one type of cell death or mixed types of cell death.

In the light of previous research, we think that PUFAs play key role in ferroptosis. In addition, it is known that anthracyclins induce different types of cell death depending on: 1) the type of cell, 2) the type of the anthracyclin, and 3) its concentration. Consequently, in a cell culture it is probable that different mechanisms coexist at the same time (Biochemical Pharmacology 1999, 57, 727–741). As we discussed in Section 4.4 (revised manuscript), our results suggest that ferroptosis might be the mechanism of cell death of HepG2 and Huh7. In the case of SNU449, our results are less compatible with ferroptosis. Probably there is a mixture of mechanisms in all cases, but ferroptosis is definitely an important mechanism for HepG2 and Huh7 cells. In addition, we now provide additional data showing the increase of a biomarker for ferroptosis (Front. Cell Dev. Biol. 2021, 9, 637162). We measured transferrin receptor in an animal model for liver cancer treated with anthracyclins (Figure 9). The increase of transferrin receptor in this model further supports our hypothesis that anthracyclins provoke ferroptosis in cell lines.

Common idea is that PUFAs are good lipids, how does the story here explain that? 

It is usually said that PUFAs are good, in particular ω-3 PUFAs by opposition to ω-6 PUFAs. However, the roles of PUFAs in the cell are complex and this might be a simplistic view (similar to “good” cholesterol and “bad” cholesterol). It is true that pioneering studies suggested the beneficial effects of ω-3 PUFAs (docosahexaenoic and eicosapentaenoic acids in oily fish) and a detrimental effect of ω-6 PUFAs (arachidonic acid, for example). After many years, many dietary interventions suggest a beneficial effect, others do not. As  dietary supplements, it appears that the first-pass metabolism in the gut and liver is not considered when their beneficial effect throughout the body is discussed. The real role is not still clear.

Our explanation of the different results is based on the fact that PUFAs are involved in different biophysicochemical processes such as membrane fluidity, ER stress, and oxidative stress by lipid peroxidation. If they are beneficial or not may depend on the cellular context and physiological/pathophysiological status (type of disease and/or even specific patient). In addition, cell mortality is not necessarily a detrimental process. In the case of non-cancerous cells, it serves to renew tissues.

Other minor comments:

  1. The data presented without statistics, just show the trend of changes.

We focused the analysis on the effect (relative increase/decrease in lipidomics) and the uncertainty associated by confidence intervals instead of p-values (or similar statistics). This strategy of data analysis has been encouraged by prestigious statisticians and mathematicians like M Gardner and D Altman (Br Med J (Clin Res Ed) 1986, 292, 746–750). The uncertainty of the fold changes is expressed in the confidence intervals. In this way, we addressed the concerns of the statistical community about the use of “statistical significance” and the apparently general misinterpretation of p-values or similar statistics (Nature 2019, 567 (7748), 305–307; Am. Stat. 2019, 73 (sup1), 1–19; Eur. J. Epidemiol. 2016, 31:337; Semin. Hematol. 2008, 45(3):135-40). 

Reviewer 2 Report

In this paper, Balgoma et al have studied changes in the lipidome of three different liver cancer cell lines induced by two anthracyclins (doxorubicin and idarubin). The authors find that the anthracyclins increase the levels of ether-containing phospholipids and free PUFAs. As a whole the major merits of the work are in lipid analysis by mass spectrometry and they are thus technical. This is a descriptive work, but the results are clear and interesting and may be of physiological relevance.

1) The title is misleading. The authors have not measured lipid peroxides or ferroptosis in their samples, so it is inappropriate to include these terms in the title. I find it somewhat surprising that the authors have not measured oxidized lipids, as they seem to possess the requisite technology and expertise. Likewise, the authors may want to consider the analysis of ferroptotic cell death in their system.

2) Lines 158-159. It is unclear what the authors are trying to say here. Please rewrite and clarify.

3) Line 207. That the authors detected only one alkyl-PC species in their cells is intriguing. Could this constitute a peculiarity of the cell lines used or a general feature of the primary liver cells from which the lines were derived?

4) In the caption to Fig. 4, the authors indicate that some of the ether phospholipids shown in the left column may actually correspond to two different structures. To avoid unnecessary confusion, this needs to be clearly indicated in the table itself, just as the authors did in the supplemental data (e.g.: write “PE(O-16:1/20:4) or PE(P-16:0/20:4)”, not just “PE(O-16:1/20:4)”.  

5) In Fig. 5, some fatty acids are repeated (FA(16:1) and FA(18:1) appear twice, FA(18:2) thrice). I understand that this may be due to the presence of positional isomers, but if the authors do not identify them (18:1n-7, 18:1n-9, etc…) , the table cannot be properly assessed.

6) Lines 288-289 and lines 343-345. I may agree that increased de novo lipid biosynthesis results in increased levels of species containing saturated and monounsaturated fatty acids, but to regard the levels of these species as indicators of LXR- and/or SREBP-dependent lipogenesis seems an oversimplification. Rapid recycling of fatty acyl chains may occur. I was wondering why the authors did not measure directly the activity of these promoters before and after treating the cells with anthracyclins. Does the amount of cellular phospholipid change after these treatments? In Fig. 2 the authors clearly show that at least one of the cell lines utilized increases in size.   

7) Discussion – general. This section is too long and highly speculative. There are whole paragraphs that are minimally based on the results presented. The authors should considerably shorten this section and discuss their own data in the context of the existing literature. While it is fine to add some speculation if this helps to clarify the biological/biomedical implication of the data or provide new directions, too much speculation detracts from the manuscript.

Author Response

We kindly thank Reviewer 2 for the comments. We have addressed them:

1) The title is misleading. The authors have not measured lipid peroxides or ferroptosis in their samples, so it is inappropriate to include these terms in the title. I find it somewhat surprising that the authors have not measured oxidized lipids, as they seem to possess the requisite technology and expertise. Likewise, the authors may want to consider the analysis of ferroptotic cell death in their system.

 We have now modified the title to a more general concept: “Anthracyclins increase PUFAs: potential implications in ER stress and cell death”

On the one hand, in a close future, we plan to perform the analysis of the products of oxidation of lipids with a targeted analytical method (triple quadrupole). Nevertheless, this was beyond the scope of this study by untargeted lipidomics. On the other hand, to reinforce the idea of ferroptosis, we have measured the transferrin receptor in a related murine model of HCC treated with doxorubicin (Figure 9 in new Section 3.8). We have discussed this figure in a brief way in lines 443-446

2) Lines 158-159. It is unclear what the authors are trying to say here. Please rewrite and clarify.

 We have rewritten how we calculated cell size (lines 158-161). We consider that this revision made it much clearer now.

3) Line 207. That the authors detected only one alkyl-PC species in their cells is intriguing. Could this constitute a peculiarity of the cell lines used or a general feature of the primary liver cells from which the lines were derived?

We think that the reason is the type of primary cells from which the cell lines derived (hepatocytes). In fact, there is a regulatory inverse correlation between the levels of synthesis of cholesterol and the levels of synthesis of alkylglycerophospholipids (FEBS Lett 2017, 591, 2720–2729). As the metabolism of the liver is very active in the synthesis of cholesterol, this may explain that the liver is relatively poor in etherPCs and hence cell lines derived from hepatocytes.

4) In the caption to Fig. 4, the authors indicate that some of the ether phospholipids shown in the left column may actually correspond to two different structures. To avoid unnecessary confusion, this needs to be clearly indicated in the table itself, just as the authors did in the supplemental data (e.g.: write “PE(O-16:1/20:4) or PE(P-16:0/20:4)”, not just “PE(O-16:1/20:4)”.  

 We have corrected Figure 4 to include both plasmanyl and plasmenyl options.

5) In Fig. 5, some fatty acids are repeated (FA(16:1) and FA(18:1) appear twice, FA(18:2) thrice). I understand that this may be due to the presence of positional isomers, but if the authors do not identify them (18:1n-7, 18:1n-9, etc…) , the table cannot be properly assessed.

To profile the lipidome in an untargeted way, we used LC-MS. This technique allows to analyze many families of lipids (mainly glycerolipids and sphingolipids), but it only allows to identify the number of carbons and unsaturations. Unfortunately, it does not allow characterizing the position of the unsaturations. As Reviewer 2 observed, we detected different isomers that were separated chromatographically and we reported them separately. However, the collision induced fragmentation of the adduct [M-H]- of FAs does not yield identificative fragments. Because of this reason, in LC-MS lipidomics, FA(20:4) is usually identified as arachidonic acid because it is the most abundant fatty acid with 20 carbons and four unsaturations. Similarly, the most abundant FA(18:1) can be safely assigned to oleic acid (18:1n-9) and FA(16:1) to palmitoleic acid (16:1n-7). We have described in materials and methods this tentative identification (lines 158-161) according to prior knowledge and the relative intensity.  We have also included it in the table of identification in Supplementary Material and in Figure 5.

6) Lines 288-289 and lines 343-345. I may agree that increased de novo lipid biosynthesis results in increased levels of species containing saturated and monounsaturated fatty acids, but to regard the levels of these species as indicators of LXR- and/or SREBP-dependent lipogenesis seems an oversimplification. Rapid recycling of fatty acyl chains may occur. I was wondering why the authors did not measure directly the activity of these promoters before and after treating the cells with anthracyclins. Does the amount of cellular phospholipid change after these treatments? In Fig. 2 the authors clearly show that at least one of the cell lines utilized increases in size.   

It is known that LXR/SREBP pathways orchestrate the de novo lipogenesis. But it is also known that they do it in a complex manner. For example, oxysterols are ligands of LXRs. But, for example, their activity is downregulated by the activation of the androgen receptor without affecting mRNA levels, degradation or DNA binding (The Journal of Biological Chemistry, 286(23), 20637–20647). Consequently, even if the mRNA levels of LXRs do not change, one cannot discard a change in the levels of the de novo lipogenesis. The lipids with saturated and/or monounsaturated fatty acids are the endpoint of the de novo lipogenesis so they reflect the activity of the pathway. In the case of the effect of the androgen receptor, they even reflect it in a better way. As Reviewer 2 states, other processes might confound our interpretation of the obtained experimental results. For example, a reduction of beta oxidation in the mitochondria could lead to the intracellular accumulation of these fatty acids. We have added this caveat in Section 4.1 (lines 361-365, revised manuscript).

Regarding cell size, we have highlighted in the discussion at Section 4.3 the potential relationship between the increase in cell size of SNU449 and the general increase in the lipidome in Figure 3. Considering Figures 4-7, the general increase in lipids does not seem associated to specific fatty acids or lipid families. We do not have an explanation for the moment. Considering the results of this non-targeted study, we plan to perform targeted experiments with isotopic labeled lipids to trace the activity of different pathways.

7) Discussion – general. This section is too long and highly speculative. There are whole paragraphs that are minimally based on the results presented. The authors should considerably shorten this section and discuss their own data in the context of the existing literature. While it is fine to add some speculation if this helps to clarify the biological/biomedical implication of the data or provide new directions, too much speculation detracts from the manuscript.

To make the discussion shorter, we have removed section 4.1. In addition, we have also reduced different parts where we discussed the results from other researchers to introduce ours. We have also removed specific speculations in sections 4.3 and 4.4. Now the discussion is more than 25% shorter (from 252 to 189 lines).

Round 2

Reviewer 2 Report

None.